# Using a Bone Conduction Hearing Device as a Tactile Aid

**Martin Kompis** \*, **Manfred Langmair, Georgios Mantokoudis, Stefan Weder, Tom Gawliczek**
and **Marco Domenico Caversaccio**

Department of ENT, Head and Neck Surgery, Inselspital, Bern University Hospital, University of Bern,
3010 Bern, Switzerland; manfred.langmair@insel.ch (M.L.); georgios.mantokoudis@insel.ch (G.M.);
stefan.weder@insel.ch (S.W.); tom.gawliczek@insel.ch (T.G.); marco.caversaccio@insel.ch (M.D.C.)
**\*** Correspondence: martin.kompis@insel.ch

**Abstract:** Background: With the advent of cochlear implants, tactile aids for the profoundly deaf
became obsolete decades ago. Nevertheless, they might still be useful in rare cases. We report the
case of a 25-year-old woman with Bosley–Salih–Alorainy Syndrome and bilateral cochlear aplasia.
Methods: After it was determined that cochlear or brainstem implants were not an option and tactile
aids were not available anymore, a bone conduction device (BCD) on a softband was tried as a tactile
aid. The usual retroauricular position and a second position close to the wrist, preferred by the patient,
were compared. Sound detection thresholds were measured with and without the aid. Additionally,
three bilaterally deaf adult cochlear implant users were tested under the same conditions. Results: At
250–1000 Hz, sounds were perceived as vibrations above approximately 45–60 dB with the device at
the wrist. Thresholds were approximately 10 dB poorer when placed retroauricularly. Differentiation
between different sounds seemed difficult. Nevertheless, the patient uses the device and can perceive
loud sounds. Conclusions: Cases where the use of tactile aids may make sense are probably very
rare. The use of BCD, placed, e.g., at the wrist, may be useful, but sound perception is limited to low
frequencies and relatively loud levels.

**Keywords:** tactile aids; vibratory sensation; cochlear aplasia; sound field; bone conduction; sound processor

## 1. Introduction

Cochlear implants have been the standard method of treatment of profound or even
severe hearing loss [1,2] for several decades, now. Their efficiency can at times be nothing
less than amazing, and cochlear implants have been called "arguably, the most successful
device at the machine-brain interface" [3].

In the light of this undeniable success, older methods and former ideas, such as tactile
aids [4–6], have become all but forgotten. However, tactile aids seem to still have been
in use as late as the beginning of this century [7]. With these devices, profoundly deaf
persons were able to perceive sounds as vibrations despite their hearing loss. Sophisticated
devices with one to seven channels or frequency bands were available [6,7]. They allowed
the vibrotactile perception of sounds, in some cases sound recognition or improved lip-
reading [6], and sometimes reportedly even limited word recognition [7]. Today, cochlear
implants are considered superior.

As the largest centre for cochlear implantation at this time in our country, we have
never used or prescribed tactile aids until, amazingly, this very year, when we learned that
they may still be of some limited value in some very rare cases.

## 2. Case Presentation

The parents of a 25-year-old woman diagnosed with Bosley–Salih–Alorainy syn-
drome [8–11] contacted us. Their daughter was completely deaf in both ears and she, as
well as her parents, wanted her to be able to perceive at least some loud sounds. Their hope

and motivation was that she might be able to react to loud warning sounds and possibly notice when somebody called out to her.

The patient had bilateral cochlear aplasia, which is known to occur in some, but not in all, patients with Bosley–Salih–Alorainy syndrome [8,11]. Figure 1 shows an MRI of the temporal bone with a bilateral cochlear aplasia, an aplasia of the labyrinth on the right and a dysplastic vestibule on the left.

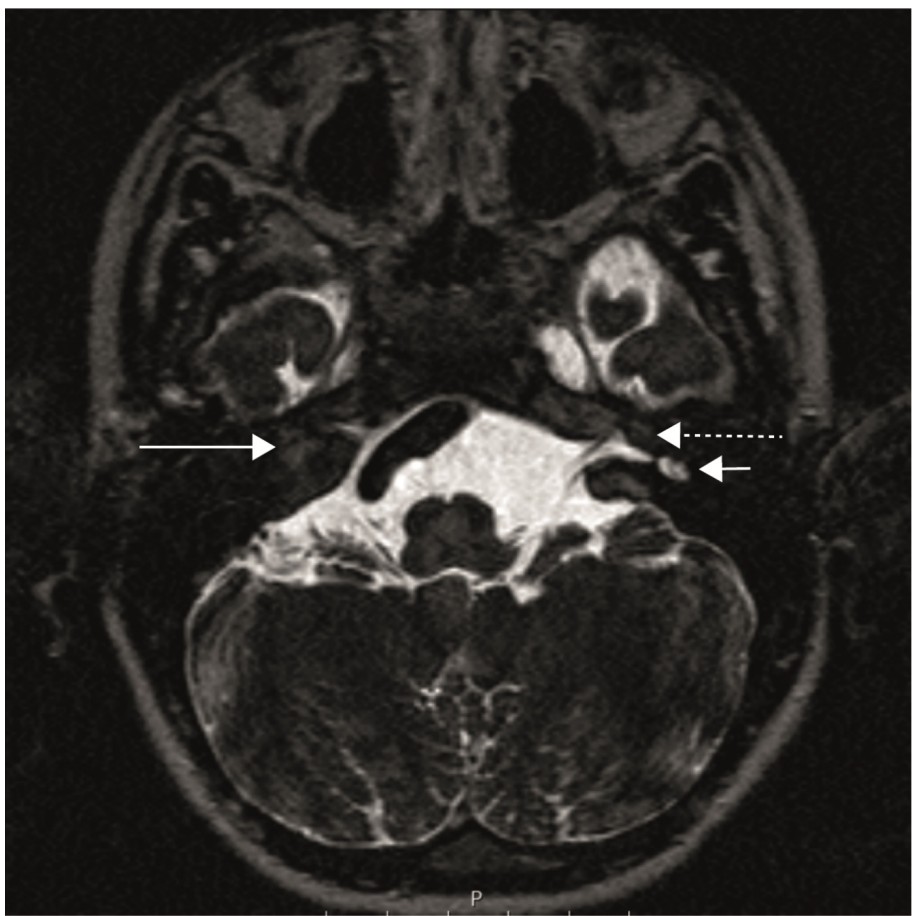

**Figure 1.** MRI of the temporal bone (axial plane), CISS sequences (constructive interference in steady state) with a right cochlear aplasia (long solid arrow), aplasia of the right inner ear canal, left cochlear aplasia (long dashed arrow) and dysplastic left vestibule (short arrow).

In the audiometric assessment, we found a bilaterally normal impedance audiometry and observed normal otoscopic findings, but no otoacoustic emissions could be evoked in either ear. In pure tone audiometry, no hearing was found in either ear, at any of the audiometric frequencies, and neither in the air conduction (AC) measurements (125–8000 Hz) nor in the bone conduction (BC) measurements (250–800 Hz) up to the audiometer limits, i.e., up to 120 dB HL for the AC measurements and up to 80 dB HL for BC measurements.

Cognitive and behavioural abnormalities have also been described in Bosley–Salih–Alorainy syndrome [9,11], and the patient showed developmental challenges and an additional steady decline of her cognitive abilities which had started approximately in her late teens. At the time of the consultations, she was able to read and to write in a limited manner. Furthermore, she exhibited a significant decrease in vertical ocular movement. Abnormalities in ocular motility are common in Bosley–Salih–Alorainy syndrome [9,10].

In the absence of both cochleae, cochlear implantation was not an option. Anatomically, a brain stem hearing implant [12,13] was conceivable, but the age of the patient, the lack of any prior hearing experience, and her continuing cognitive decline led to the decision against this route. In this decision-making process, our prior experiences with auditory

brainstem implants and with poor results after late cochlear implantations in congenitally deaf adults and the cognitive decline played an important role.

With the options thus severely limited, we searched for tactile aids but did not find any that were commercially available anymore. In this situation, we performed a trial with a Cochlear Baha 6 max (Cochlear Inc., Mölnlycke, Sweden) [14] bone conduction (BC) sound processor fixed on a softband [15]. At higher sound levels and at low frequencies, its vibrations can be easily felt with the fingertips. Nevertheless, we had never previously tried to use it in other positions than that mounted on the head, and we had never seriously considered these vibrations to be a possibly useful output signal for a user. Before the actual trial, we evaluated other possible bone conduction devices, namely the ADHEAR (Medel Inc., Innsbruck, Austria) and the Ponto 5 SuperPower Device (Oticon Medical, Askim, Sweden). With the Baha 6 max, we believed we had found a reasonable compromise between size, weight and attainable output force levels in the low frequency range, which is important for this application. Nevertheless, we believe that it is very much possible that these other devices are similarly well suited.

We tried the sound processor on a softband in the usual position behind the ear, and indeed the patient could perceive loud sounds in the order of magnitude of 70 dB to 90 dB HL in the frequency range of 250 to 1000 Hz as vibrations. In contrast, no sound detection was found without the device, at least not up to the maximum levels available with our sound field audiometer. The audiometer was an Equinox 2.0 (Interacoustics A/S, Middelfart, Denmark) and it was connected to a Genelec type 8030C loudspeaker (Iisalmi, Finland). The frequency-dependent maximum sound field levels of the system are shown in Figure 2. The device was fitted using the Cochlear Fitting Suite 1.10.22628.0 (Cochlear Inc., Mölnlycke, Sweden) in such a way as to have a high gain at the lower frequencies below 2 kHz. The gain was limited above 2 kHz, as no vibratory sensation was found at these higher frequencies and acoustical feedback could thus be limited.

**Figure 2.** Sound field measurement with the bone conduction aid used as a tactile aid in the classic retro auricular position (blue squares) and on the hand (red circles) in the reported patient (thick lines and filled symbols) and 3 profoundly deaf controls (thin lines and empty symbols).

We felt that perception thresholds as high as 70 to 90 dB HL were not satisfactory. In order to improve the range, we tried another position: above the wrist of the subject (Figure 3). Indeed, perception thresholds improved by 15 to 25 dB, as shown in Figure 2.

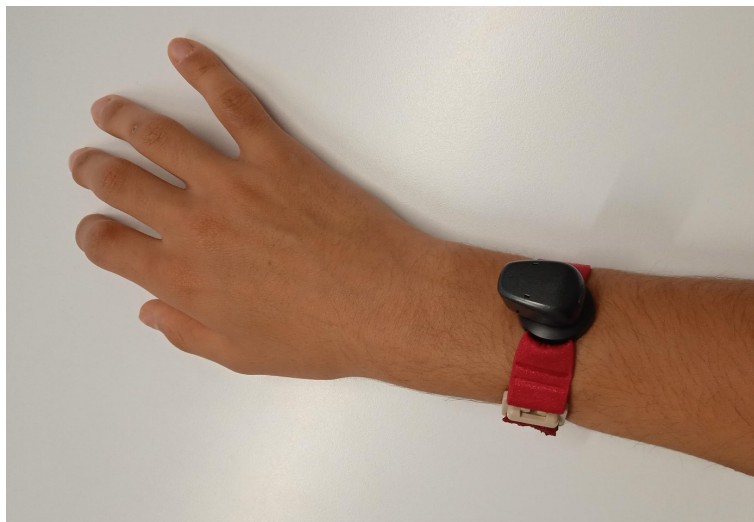

**Figure 3.** A bone conduction device (Cochlear Baha 6 max) used as a tactile aid just behind the wrist on a shortened softband.

The patient took the device home for a trial and now uses it on an irregular, almost but not quite daily basis, for up to several hours a day. She clearly prefers the position close to the wrist over any placement on the head. For practical reasons (limitation of the movement of her hand), she prefers a position behind the wrist, as shown in Figure 3.

## 3. Comparative Measurements

As we had no prior experience of our own, it was unclear to us whether the absolute perception thresholds found were typical and, specifically, if the difference between the head and the hand positions was real and could also be found in other subjects. A literature search was rather unfruitful, as research on vibratory sensations seems to be mostly concentrated on low to very low frequencies of often 300 Hz or even lower, and to sensations at the hands, e.g., [16].

In order to learn more about these vibratory thresholds, we asked three cochlear implant users to help us with a limited additional evaluation. Two of the subjects were male, one was female. The age range was 23–85 years. All three had bilateral profound deafness and air conduction hearing thresholds well above the maximum sound field levels our audiometer could emit at any frequency. All of their bone-conduction hearing thresholds were above the maximum output limits of our audiometer, but one control subject was able to perceive the vibration of the bone vibrator at 250 Hz and 500 Hz, as shown in Figure 4. The sensation he described was clearly tactile and not hearing.

Their perception thresholds were measured under sound-field conditions using narrow-band noise signals, with their cochlear implant sound processors taken off. Measurements were performed with the BC sound processor mounted on a softband and placed either in the usual position behind the ear on their non-implanted side, or the wrist, in a position as similar as possible to the one preferred by the patient. Vibratory sensation rather than auditory perceptions were expected and also reported by all three subjects. For all subjects, a third measurement was performed without any sound processor.

Figure 2 shows the results. None of the subjects could perceive any of the sounds presented without their cochlear implant processor and without the BC sound processor. With the BC device in place, thresholds were lowest (best) in the frequency range 250 to 750 Hz and the average difference between the head and the hand positions was found to be exactly 10 dB in our small sample. However, in one subject, thresholds were slightly better behind the ear above 750 Hz.

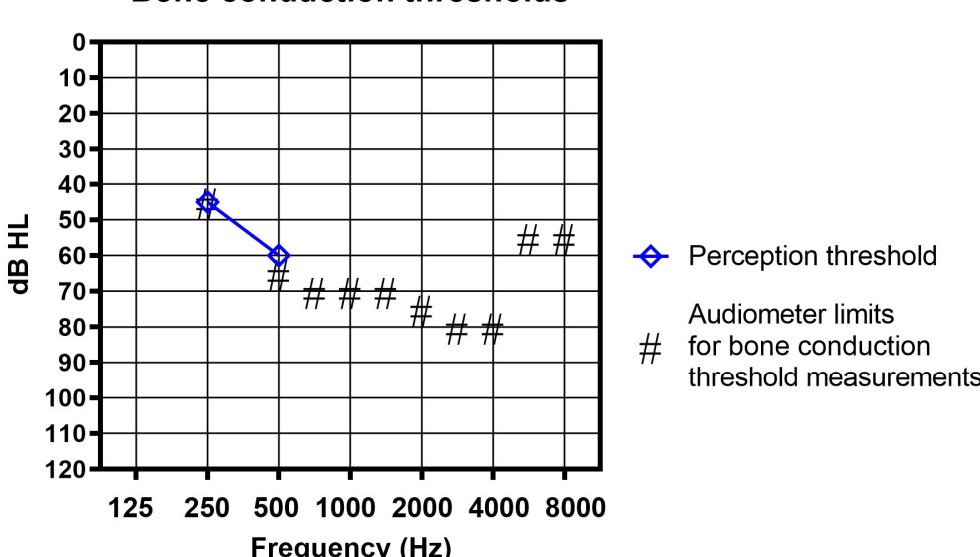

**Figure 4.** Bone conduction (BC) threshold measurements. Neither the patient nor any of the controls reported any hearing sensation in the conventional BC thresholds measurement up to the frequency-dependent maximum levels (#) available with our audiometers. One control subject reliably detected the vibration at 250 and at 500 Hz (shown here as blue diamonds) and described the sensation clearly as tactile and not as hearing.

Above the individual thresholds, sounds could be reliably detected and very rough temporal patterns (basically switching the acoustic signal on and off again) were reliably perceived. Any further discrimination seemed to be next to impossible, at least without training.

## 4. Discussion

Cochlear implants have been used since 1990 at our department and, until recently, the use of tactile aids was not even seriously considered. Somewhat to our surprise, we found that there may still be justifiable applications, although they are probably very rare. It is interesting to note that a very different application of tactile aids than the one reported here may also be useful. Specifically, it has been found that a multisensory approach, i.e., combined auditory and tactile stimulation, can improve speech understanding in noise [17].

The bone conduction sound processor used as a tactile aid does indeed allow the perception of acoustical signals. However, it is limited to relatively loud sounds and to low frequencies, and the sensation is vibrotactile. By itself, it is certainly unsuitable for any but possibly the simplest forms of oral communication. It is unclear how much it can help, e.g., by supporting lip reading, as the limited cognitive capabilities of our patient did not allow a closer examination. We would expect at most a limited help. Nevertheless, the patient does use the device and it seems to be helpful to detect some of the louder acoustical signals in her surroundings.

Bone conduction devices such as the one used here were developed to elicit auditory sensations via the BC pathway, the cochlea and, ultimately, the auditory nerve. We have no indication that this is happening here. All subjects reported clearly tactile sensations and no hearing sensations through the BC device. This holds true for the position at the head as well as at the wrist. Consequently, we believe that we have measured purely tactile sensations.

The placement of the BC processor does affect perception thresholds. The placement behind the wrist seems, in this respect, to be better than the normal placement of BC hearing aids behind the ear, not only in our patient, but mostly also in the small sample tested. Certainly, each placement has also its own practical challenges, such as, e.g., limitations

of own movements, visibility, or risk of contact to clothing. It is conceivable that other placements than the two reported here may be better.

Although BC devices can be used as tactile aids to perceive acoustic signals, it is important to stress that in most cases this is one of the last resorts. If at least one cochlea with an intact auditory nerve is present, cochlear implantation should clearly be considered first and will probably lead to much better results in most cases. Even in patients with bilateral cochlear aplasia and bilateral auditory nerve aplasia, the use of tactile aids is one of the last solutions to be evaluated. Auditory brainstem implants should be considered first, even though results are generally poorer than with cochlear implants [12]. As with all auditory implants in congenitally and bilaterally deaf patients, early implantation is a key factor for its success.

## 5. Conclusions

Cases where the use of tactile aids may make sense are probably very rare. The use of bone conduction devices, placed preferably close to the wrist, may be useful, although the perception of sound signals is limited to low frequencies and relatively loud levels, and sound discrimination must be expected to be very poor.

**Author Contributions:** Conceptualization, M.K., G.M. and S.W.; Methodology, M.K.; Validation, M.K., M.L. and T.G.; Formal Analysis, M.K. and G.M.; Investigation, M.K. and M.L.; Resources, M.D.C. and M.K.; Writing—Original Draft Preparation, M.K.; Writing—Review and Editing, M.K., G.M., S.W., M.D.C. and T.G.; Visualization, M.K. and G.M.; Supervision, M.K. and M.D.C.; Project Administration, M.K. All authors have read and agreed to the published version of the manuscript.

**Funding:** This research received no external funding.

**Institutional Review Board Statement:** Ethical review and approval were waived for this study, as no such board reviews are available for case reports in our country.

**Informed Consent Statement:** Informed consent was obtained in writing from all subjects involved in this report, including the patient, her parents and the cochlear implant users.

**Data Availability Statement:** Not applicable.

**Conflicts of Interest:** The authors declare no conflict of interest.

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
