# Peer review of "Using a Bone Conduction Hearing Device as a Tactile Aid"

_audiolres, doi:10.3390/audiolres13030040_

Round 1

Reviewer 1 Report

Dear Editor

After reading the manuscript titled "Using Bone Conduction Hearing Devices as a tactile Aid" submitted to: Audiology Research, that brings a case report with bilateral aplasia, in which a BAHA was used as a tactile aid. I suggest that the article is suitable for publication, after several changes are required before acceptance for publication.

The paper describes a case with bilateral aplasia, and since tactile aids are no more available, they made use of a BAHA as a tactile aid. The paper lacks an explanation of how the BAHA excited the sensory pathway. The authors should discuss the issue of whether the sensory nerve involved is the auditory nerve or tactile sensation. If they think it is auditory, then they should try and explain how the stimuli at the wrist reach, and excite the auditory system. There is need to elaborate why the wrist was found to give better thresholds compared to the mastoid, which pathways were activated.

I would guess the somatosensory system is more sensitive in the hand, and if the physiology is activation of a tactile pathway rather than the hearing pathway via 8th nerve – this should be further explained.

Others have attempted to make use of tactile aids in patients with hearing loss. For example-Cieśla K, Wolak T, Lorens A, Heimler B, Skarżyński H, Amedi A. Immediate improvement of speech-in-noise perception through multisensory stimulation via an auditory to tactile sensory substitution. Restor Neurol Neurosci. 2019;37(2):155-166. doi: 10.3233/RNN-190898. PMID: 31006700; PMCID: PMC6598101.

Minor changes suggested:

Line 10- 25-year old women with- woman

Line 45-ocular motility-should be explained.

L 48-49-abilities stating approximately-should be-starting or beginning

61 & 78-Baha-perhaps better-BAHA

L64- positions that mounted on-should be-than that or other than

L69,70 with our sound field audiometer.-explain

L71-high gain in the lower-at the lower

74-that perceptions thresholds- should be perception thresholds

L75,78-another position behind the wrist-better-above the wrist or beyond the wrist

81- as a tactile aids in should be- aid

82-  (broad  lines-should be-thick lines

87-a position rather far back on the forearm; fig text says-behind the wrist-be consistent.

91-positions were real-should be-was

91-in other persons-better-subjects.

97-sound filed levels should be-field

98-subject -should be- subjects

109-thresholds were lowest (best) were in the frequency range-there is an extra  “were” here.

113-could be reliable detected-should be-reliably

127-e louder acoustical signal in her should be-signals

Reviewer 2 Report

N/A

The authors may need one more check on the whole manuscript for the grammar and use of words. The content is fine.

Reviewer 3 Report

This case report presented the availability of BCD as a tactile aid for one case with cochleae aplasia. The position to put the devise on the wrist may be better than mastoid. 

 Comment #1: It would be more helpful to understand the hearing loss of this patient if you show us the more detail of audiometric tests such as bone conduction threshold and ABR. Was this patient complete deaf as in line 38?

Comment #2: The description in line 15 “Results: At 250-1000 Hz, sounds were perceived above approximately 45-60 dB with the device at the wrist” and in line 67-69 “indeed the patient could perceive loud sounds in the order of magnitude of 70 dB to 90 dB HL in the frequency range of 250 to 1000 Hz” was not consistent with complete deaf. You should distinguish and describe more precisely between vibration threshold and sound threshold.

Comment #3: For the control cases, authors compared the threshold between mastoid and wrist. The bone conduction thresholds of sound for the conventional audiometry should be shown.

Reviewer 4 Report

The authors describe a rare case of 25-year old women with Bosley-Salih-Alorainy-Syndrome and bilateral cochlear aplasia.  A bone conduction device on a softband was tried as a tactile aid. 

Can the authors please elaborate a little bit more in detail as of why an ABI was not considered? Age and the lack of prior hearing experience, hence not surprising the cognitive decline seems not enough. Especially since ABI application shows great results and benefit for patients.

Furthermore is the patient using this set-up now on a daily basis (to what extent hours/day?)  

Were other BCD devices considered/tested? 

Spelling and English check is recommended for example cognitive is misspelled etc...

Reviewer 5 Report

The authors wrote a case report regarding a Bone Conduction Hearing Device used as a Tactile Aid in a patient with Bosley-Salih-Alorainy-Syndrome.

The condition is very rare and the case is very interesting and well written.

Please follow these suggestions to improve the quality of article and give a better impact in audiological literature.

1. In the introduction, please talk more about the tactile aids.

2. In the discussion please talk more about the possible solution in cases of cochlear aplasia or auditory nerve aplasia, (ABI etc...)

Round 2

Reviewer 3 Report

Authors corrected the manuscript suitably. I think this revised manuscript is suitable for the acceptance.